# FLCatcher: Fingerprinting Poisoning Attack in Non-IID Federated Learning

## Abstract

Currently, extensive efforts have been made to defend against poisoning attacks in Federated Learning (FL). However, most existing defenses fall short in a more general and practical scenario, i.e., Non-IID FL. The core problem of current defenses lies in the fact that they all basically identify poisoned gradients by observing the inter-client gradient distribution difference. However, the inherent data heterogeneity in Non-IID FL naturally induces such gradient variations, rendering malicious gradients indistinguishable from benign ones. To address this, we propose FLCatcher, a novel defense framework tailored to Non-IID poisoning attacks from a two-perspective consideration. First, we observe that despite the data heterogeneity of Non-IID FL, the gradient evolution trajectory of benign clients tends to follow a consistent direction over time, whereas malicious clients persistently generate gradients deviating from expected trajectories to degrade the global model. Leveraging this insight, FLCatcher designs an adaptive discriminative gap amplification mechanism, which dynamically calibrates per-client detection thresholds by tracking long-term behavioral biases. Second, FLCatcher proposes a Wasserstein distance-based distributional alignment strategy to quantify subtle, layer-wise gradient deviations, enabling the identification of malicious perturbations that may be obscured within normal client variability. Extensive experiments on standard FL benchmarks evaluate the effectiveness of FLCatcher. Specifically, under Non-IID settings, FLCatcher achieves an average TPR exceeding 94.47% and an average FPR below 0.72%, significantly outperforming state-of-the-art defenses.

## 1 Introduction

Federated Learning (FL) (Pei et al., 2024) has emerged as a promising paradigm for distributed machine learning, enabling multiple clients to collaboratively train a global model without exposing their local data. In practice, client data in FL is often Non-Independent and Identically Distributed (Non-IID), due to user-specific behaviors, diverse sensing conditions, and task-driven data generation (Lu et al., 2024). This intrinsic statistical heterogeneity not only complicates global model convergence but also introduces significant security vulnerabilities—especially to poisoning attacks.

Poisoning attacks (Khuu et al., 2024) exploit the inherent vulnerabilities of FL (Martínez Beltrán et al., 2023; Beltrán et al., 2023), particularly its decentralized nature and lack of visibility into the local training, to compromise model integrity. In poisoning attacks, adversaries intentionally manipulate local data or craft malicious updates, often disguising them to resemble benign ones in both direction and magnitude (Krauß et al., 2024; Ma et al., 2025), thereby circumventing conventional similarity-based or norm-based aggregation rules in FL (Fung et al., 2020). These threats are further amplified by the emergence of adaptive poisoning attacks (Yang et al., 2024; Zhang & Huang, 2024), where adversaries carefully tune the direction, magnitude, and timing of malicious updates to mimic benign client behavior. Such sophisticated evasion strategies significantly undermine existing defense techniques (Zhang et al., 2022; Cao et al., 2020; Yin et al., 2018) that rely on static heuristics or single-round observations, posing serious threats to FL. Moreover, the inherent Non-IID nature of FL introduces significant statistical variability across client updates, making it increasingly difficult to distinguish between malicious and honest behavior.

*From the attacker's perspective*, Non-IID heterogeneity offers a natural camouflage. Malicious clients can exploit the inherent variability in client updates to perform slow-drift or distribution-mimicking attacks, in which poisoned gradients closely imitate the update patterns of honest clients (Liu et al., 2024; Rong et al., 2022). By gradually deviating from historical gradients or imitating the distributional patterns of honest clients, attackers can insert backdoors or corrupt the global model without triggering outlier detection (Liu et al., 2024; Rong et al., 2022). More advanced threats include adaptive poisoning attacks (Fang et al., 2020) or Sybil attacks (Werthenbach & Pouwelse, 2023), where multiple colluding adversaries submit aligned gradients consistent with the benign statistical noise, further challenging existing defenses.

*From the defender's perspective*, Non-IID heterogeneity poses significant challenges to existing defenses. Existing aggregation rules, such as RFA (Pillutla et al., 2022), and Norm Clipping (McMahan et al., 2017)—rely on basic geometric or norm-based metrics that assume benign updates cluster closely in the parameter space. However, Non-IID data causes honest clients to produce diverse gradients, blurring the distinction between benign and malicious updates. Similarity-based defenses like FoolsGold (Fung et al., 2020) and FLTrust (Cao et al., 2020), which assume independence among clients or rely on similarity scoring, become fragile when adversaries coordinate or manipulate trust metrics. Methods such as Trmean (Yin et al., 2018), Bulyan (Guerraoui et al., 2018), and Mkrum (Blanchard et al., 2017) require fixed assumptions about attacker ratios or hand-tuned hyperparameters, which limit their adaptability to adaptive attacks that subtly adjust perturbation strength. The existing methods, FLDectector (Zhang et al., 2022) and FedRecover (Cao et al., 2023), assume a constant distribution of client data. However, in Non-IID scenarios, the data may change at any time, which can lead to misjudgments and a lack of monitoring of client-qua-round consistency (Zhang et al., 2024). Existing defense strategies predominantly rely on inter-client gradient distribution analysis to identify outliers. Yet, in Non-IID FL, benign gradient updates inherently exhibit significant variability due to divergent data distributions. This overlap between natural and malicious deviations renders many current defenses ineffective, as poisoned updates can easily be masked by natural gradient fluctuations.

**Our contributions.** To tackle these challenges, we propose FLCatcher, a novel federated behavior embedding framework that robustly distinguishes between benign heterogeneity and malicious perturbations by jointly modeling the spatial distribution and temporal evolution of client gradients. The main contributions are summarized as follows:

- **Wasserstein Distance-based Distributional Alignment:** To capture subtle, layer-wise deviations, FLCatcher employs a Wasserstein distance metric to measure distributional shifts across client gradients, effectively identifying malicious perturbations even when they are camouflaged within natural variability.

- **Adaptive Discriminative Gap Amplification** : By analyzing gradient evolution trajectories, we observe that benign clients tend to follow consistent update directions over time, while adversarial clients generate persistent deviations. FLCatcher exploits this insight through an adaptive discriminative gap amplification mechanism, which dynamically adjusts per-client detection thresholds based on long-term behavioral patterns.

- **Superior Performance in Non-IID FL.** We conducted extensive experiments on standard FL benchmarks, demonstrating that FLCatcher consistently achieves superior detection performance and robustness. Under Non-IID settings, FLCatcher attains an average TPR exceeding 94.47% and an average FPR below 0.72%, significantly outperforming state-of-the-art methods.

## 2 PROBLEM FORMULATION

### 2.1 THREAT MODEL.

We consider a *model poisoning adversary* that controls a subset $\mathcal{M} \subseteq \{1, \ldots, N\}$ of clients in FL. The adversary's objective is to degrade the performance of the global model $\mathbf{W}$ while evading detection by robust aggregation methods such as Krum or RFA. The adversary has full control over the local training process and can arbitrarily manipulate the model updates $\{\Delta \mathbf{w}_i\}_{i \in \mathcal{M}}$ submitted by the compromised clients. Model poisoning attacks can generally be categorized as follows:

*Non-adaptive attacks:* In these attacks, the adversary either modifies local training data or directly crafts malicious gradients or updates without dynamically reacting to the aggregation process. Examples include label-flipping, which flips training labels by a fixed offset to mislead model training (Han et al., 2012), and the LIE attack, where the attacker adds a small fixed perturbation to the average benign gradient to evade detection (Gilad et al., 2019).

*Adaptive attacks:* These attacks balance stealthiness and effectiveness by specifically targeting robust aggregation defenses under non-IID data distributions; representative methods include the Fang attack, which crafts scaled perturbations opposing benign gradients to bypass robust aggregation (Fang et al., 2020), the Min-Max and Min-Sum attacks that respectively maximize perturbation while keeping malicious gradients close to benign ones and minimize the total distance to benign gradients to pass aggregation filters (Shejwalkar & Houmansadr, 2021), and FedGhost, which generates data-free, stealthy malicious updates that maximize attack impact while minimizing variance (Ma et al., 2025).

## 2.2 DESIGN GOALS.

In FLCatcher, we consider a standard FL system, specifically following the FedAvg protocol, consisting of a central server and a set of $\mathcal{C} = \{c_1, c_2, \ldots, c_N\}$. Each client $c_i$ holds a private local dataset $\mathcal{D}_i$ and participates in collaborative model training without sharing its raw data. The objective of FLCatcher is to enhance the robustness of Non-IID FL under model poisoning attacks, including:

*Robustness to Non-IID Heterogeneity:* We aim to effectively handle the inherent statistical heterogeneity of client data distributions. This heterogeneity often causes benign gradients to exhibit significant variance, making it challenging to differentiate natural variations from adversarial perturbations. FLCatcher must therefore minimize both false positives (classifying benign updates as malicious) and false negatives (missing actual malicious updates).

*Effective Malicious Update Mitigation:* We aim to design a defense strategy that can accurately identify and mitigate malicious updates. This includes scenarios where attackers adaptively craft poisoned gradients to mimic benign behavior, especially in complex Non-IID settings where benign client updates naturally vary significantly.

## 3 METHODOLOGY

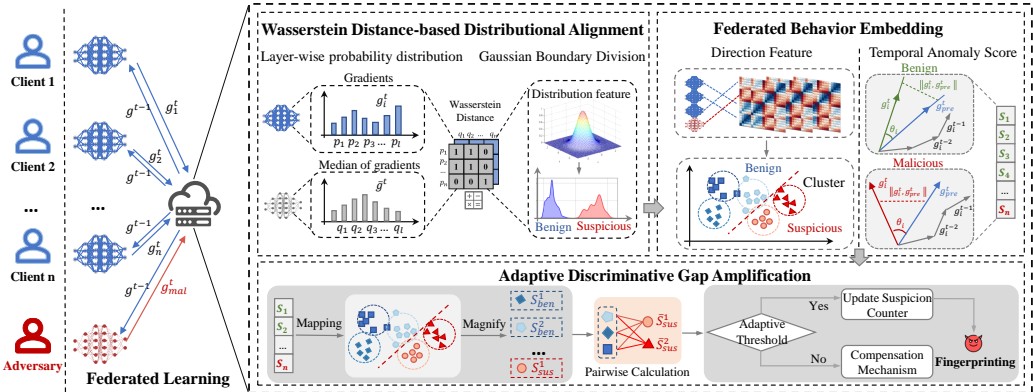

Figure 1: FLCatcher Framework.

In FL, the distributed nature and the intrinsic heterogeneity of client data make FL highly vulnerable to model poisoning attacks, especially in Non-IID environments. In such settings, benign client gradients naturally exhibit significant variance, which blurs the boundary between benign and malicious updates. Adversaries exploit this heterogeneity by crafting poisoned updates that mimic extreme benign behavior, effectively evading existing robust aggregation mechanisms. To address

these challenges, FLCatcher proposes a federated behavior embedding fingerprinting framework (See Figure 1), achieving high-precision detection of unknown attacks with federated behavior embedding and adaptive discriminative gap amplification.

- *Federated Behavior Embedding*: FLCatcher embeds static distribution alignment and dynamic temporal evolution mechanisms to capture spatial distribution and temporal evolution features of gradients.
- *Adaptive Discriminative Gap Amplification*: We design a prior-free dynamic threshold strategy that adaptively amplifies the gap between benign and malicious behaviors.

## 3.1 CONSTRUCTION OF FLCATCHER

### 3.1.1 WASSERSTEIN DISTANCE-BASED DISTRIBUTIONAL ALIGNMENT.

Existing defense methods identify malicious gradients by analyzing the overall direction or numerical differences of the gradient. However, the combination of the spatial location of the gradient and the corresponding numerical values can truly determine the correctness of the model update.

To distinguish malicious clients from benign ones with extreme Non-IID behaviors, FLCatcher first quantifies each client's distributional deviation from the median of all clients. Let the gradient of $i$-th client at round $t$ be $g_i^t = [f_i^{(1)}, f_i^{(2)}, \ldots, f_i^{(L)}]$, where $f_i^{(l)}$ is the gradient vector at layer $l$. We model $f_i^{(l)}$ as a probability distribution: $\mu_i^{(l)} = (\alpha^{(l)}, X_i[l])$, where $X_i[l]$ denotes the support (e.g., neuron gradients), and $\alpha^{(l)}$ is the weighting (initially uniform or importance-weighted). For each layer $l$, we compute the Wasserstein distance between $i$-th client's gradient distribution and the median distribution $\nu^{(l)}$:

$$\mathcal{W}_i^{(l)} = OT(\mu_i^{(l)}, \nu^{(l)}, D_S),$$

where $OT$ denotes the optimal transport solver and $D_S$ is the ground distance metric. We aggregate the distances across layers to obtain the total deviation score:

$$d_i^t = \sum_{l=1}^{L} \alpha_l \cdot \mathcal{W}_i^{(l)},$$

where $\alpha_l$ is the layer-specific weight.

Nietert et al. (2023) performs robust distribution estimation at the Wasserstein distance, which allows a certain proportion of outliers to be removed from the observation distribution to reduce the impact of malicious noise pollution.So we assume benign deviation scores approximately follow a Gaussian distribution and estimate the mean $\mu_d$ and standard deviation $\sigma_d$ of $d_i$ at each round. Clients with $d_i^t > \mu_d + k\sigma_d$ are flagged as suspicious.

To further refine detection under Non-IID conditions, we calculate the cosine similarity between each client pair to reduce the dimension and sharpen its gradient characteristics. then cluster benign clients and suspicious clients using DBSCAN to capture semantic consistency and differentiate truly malicious updates from benign extremes. As a result, we obtain benign clusters $\mathcal{C}_b$ and suspicious clusters $\mathcal{C}_s$, which serves the adaptive discriminative gap amplification mechanism.

### 3.1.2 ADAPTIVE DISCRIMINATIVE GAP AMPLIFICATION

To eliminate reliance on pre-defined thresholds or attacker ratio assumptions, FLCatcher designs a prior-free and dynamic thresholding mechanism, which amplifies the behavioral gap between benign and malicious clusters by jointly considering temporal stability and distributional deviations.

*i) Federated Behavior Embedding.* FLCatcher constructs a joint behavior embedding for each client by combining: a temporal anomaly score that captures deviations over time, and a static distributional deviation score that reflects instantaneous discrepancies. The temporal score is combined with the static distributional deviation score to form a joint behavior embedding for each client.

FLCatcher incorporates a dynamic layer that models the temporal evolution of client updates using Exponential Moving Average (EMA):

$$\hat{g}_i^t = \beta\hat{g}_i^{t-1} + (1 - \beta)g_i^{t-1},$$

where $\hat{g}_i^t$ is the predicted gradient and $\beta$ is the decay factor.

The temporal anomaly score is defined as $S_i^t$, which quantifies the deviation between $g_i^t$ and $\hat{g}_i^t$. Benign clients typically exhibit low $S_i^t$ values because their update patterns remain stable over time (Wang et al., 2025). In contrast, malicious clients while adapting their attack strategies tend to produce higher deviations from the EMA predicted gradients, resulting in larger $S_i^t$. To further amplify the distinction between malicious and Non-IID clients, FLCatcher aggregates scores at the cluster level:

$$\bar{S}_b^{(k_1)} = \frac{1}{|\mathcal{C}_b^{(k_1)}|} \sum_{i \in \mathcal{C}_b^{(k_1)}} S_i^t, \quad \bar{S}_s^{(k_2)} = \frac{1}{|\mathcal{C}_s^{(k_2)}|} \sum_{j \in \mathcal{C}_s^{(k_2)}} S_j^t.$$

*ii) Dynamic Fluctuation Score.* For each suspicious cluster $\mathcal{C}_s^{(k_2)}$, a fluctuation score $F^{(k_2)}$ is computed by quantifying its stability deviation from all benign clusters $\mathcal{C}_b^{(k_1)}$ ($k_1 = 1, 2, \ldots, K_b$), where $K_b$ is the number of benign clusters:

$$F^{(k_2)} = \frac{1}{K_b} \sum_{k_1=1}^{K_b} \left| \bar{S}_s^{(k_2)} - \bar{S}_b^{(k_1)} \right|,$$

where $\bar{S}_s^{(k_2)}$ and $\bar{S}_b^{(k_1)}$ denote the average temporal stability scores of the $k_2$-th suspicious cluster and $k_1$-th benign cluster, respectively.

*iii) Adaptive Threshold.* A suspicious cluster $\mathcal{C}_s^{(k_2)}$ is classified as malicious if:

$$\mathcal{C}_s^{(k_2)} \in \text{Malicious} \quad \text{if} \quad F^{(k_2)} > \tau,$$

where $\tau$ is a dynamically learned threshold updated from the empirical distribution of fluctuation scores $F^{(k_2)}$ over historical rounds. To mitigate false positives from natural gradient volatility under Non-IID settings, FLCatcher maintains a client-mark queue that accumulates suspicion counts, flagging only clients with persistent abnormality as malicious. The details are shown in Algorithm 1.

---

**Algorithm 1** Adaptive Threshold Decision (ATD)

---

**Input:** Cluster dictionary $\mathcal{D}_{ben/sus} = \{(L_k, S_k, \mathcal{I}_k) \mid k = 1, 2, \ldots, K\}$, suspicion counter $\mathcal{C}_s$.
**Output:** Updated suspicion counter $\mathcal{C}_s$.
  1: Identify most anomalous suspicious cluster:
     $(L^*, S^*) \leftarrow \arg\max_{(L,S) \in \mathcal{D}_{sus}} S$
  2: Compute adaptive thresholds:
     $\Delta_1 \leftarrow S^* - \max(\mathcal{D}_{ben})$                           *// gap between top suspicious and benign max*
     $\Delta_2 \leftarrow \max(\mathcal{D}_{ben}) - \min(\mathcal{D}_{ben})$                      *// benign range*
  3: **if** $l^* \neq -1$ **and** $\Delta_1 > \max(\Delta_2, R)$ **then**
  4:     Update dynamic range: $R \leftarrow \max(\Delta_2, R)$
  5:     Identify suspicious clients:
     $\mathcal{I}_{sus} \leftarrow \{C_i \mid L_i = L^*\}$
  6:     Increment suspicion count for each suspicious client:
     $\mathcal{C}_s \mathrel{+}= 1 \quad \forall j \in \mathcal{I}_{sus}$
  7: **end if**
  8: **return** $\mathcal{C}_s$

---

# 4 EXPERIMENTS

We conduct extensive experiments to evaluate the effectiveness of FLCatcher under both IID and Non-IID settings.

## 4.1 EXPERIMENTAL SETUP

**Datasets and Models.** We evaluate FLCatcher on three widely-used image classification datasets: MNIST, CIFAR-10, and Fashion-MNIST. For CIFAR-10, we adopt AlexNet as the backbone model, while for MNIST and Fashion-MNIST we use a multi-layer perceptron (MLPNet) as the backbone.

| Dataset | Attack | Mkrum | | Bulyan | | DnC | | Ours | |
|---------|--------|-------|-------|--------|-------|------|------|------|------|
| | | TPR | FPR | TPR | FPR | TPR | FPR | TPR | FPR |
| CIFAR-10 (AlexNet) | LIE | 0.007 | 0.5483 | 0.0472 | 0.4882 | 0.0003 | 0.2499 | **0.9867** | **0** |
| | Label-Flip | 0.4957 | 0.4261 | 0.5585 | 0.3604 | 0.9435 | 0.0141 | **0.9635** | **0** |
| | Fang | 0.7146 | 0.3713 | 0.0495 | 0.4876 | 0.2997 | 0.4251 | **0.9867** | 0.1292 |
| | Min-Max | 0.6409 | 0.3898 | 0.4312 | 0.3922 | **0.9900** | 0.0025 | 0.9834 | **0** |
| | Min-Sum | 0.0003 | 0.5499 | 0.0000 | 0.5000 | 0.0000 | 0.2500 | **0.9867** | **0** |
| | FedGhost | 0.6409 | 0.3898 | 0.3252 | 0.4187 | 0.4449 | 0.1388 | **0.8808** | **0** |
| MNIST (MLPNet) | LIE | 0.0608 | 0.5348 | 0.0223 | 0.4944 | 0.0003 | 0.2499 | **0.9893** | **0** |
| | Label-Flip | **0.9701** | 0.3075 | 0.9372 | 0.2657 | 0.8256 | 0.0436 | 0.8641 | **0** |
| | Fang | 0.0120 | 0.5470 | 0.0193 | 0.4952 | 0.2262 | 0.4434 | **0.9643** | **0** |
| | Min-Max | 0.8339 | 0.3415 | 0.5508 | 0.3623 | 0.9967 | 0.0008 | **0.9357** | **0** |
| | Min-Sum | 0.4096 | 0.4476 | 0.2648 | 0.4338 | 0.0179 | 0.2455 | **0.9250** | **0** |
| | FedGhost | 0.3930 | 0.4517 | 0.3279 | 0.4180 | 0.5186 | 0.1203 | **0.8779** | **0** |
| Fashion-MNIST (MLPNet) | LIE | 0.1296 | 0.5176 | 0.0784 | 0.4804 | 0.0073 | 0.2482 | **0.9867** | **0** |
| | Label-Flip | **0.9811** | 0.3047 | 0.9329 | 0.2668 | 0.1771 | 0.2057 | 0.8704 | **0** |
| | Fang | 0.0100 | 0.5475 | 0.0435 | 0.4891 | 0.2359 | 0.4410 | **0.9559** | **0** |
| | Min-Max | 0.9375 | 0.3156 | 0.9066 | 0.2733 | **0.9980** | 0.0005 | 0.9834 | **0** |
| | Min-Sum | 0.7535 | 0.3616 | 0.8814 | 0.2797 | 0.0000 | 0.2500 | **0.9668** | **0** |
| | FedGhost | 0.5664 | 0.4084 | 0.5037 | 0.3741 | 0.2818 | 0.1795 | **0.8979** | **0** |
| Average | | 0.4754 | 0.4312 | 0.3822 | 0.4044 | 0.3869 | 0.1949 | **0.9447** | **0.0072** |

Table 1: TPR and FPR comparison of different defense methods under various attacks across datasets.

**FL Setup.** By default, we consider $N = 50$ clients, of which $20\%$ (i.e., *attacker* = 10) are malicious and the effect of attack ratio ($r$) is compared for $r = [0.1, 0.2, 0.3, 0.4]$. For each dataset, we reserve $20\%$ of the training samples as each client's local validation set. Each client trains its local model using stochastic gradient descent (SGD) with a batch size of 125 per communication round. Following prior works, the learning rate is set to $0.1$ for MNIST and Fashion-MNIST, and to $0.5$ for CIFAR-10, with exponential decay at a rate of $c$. We run FL for 800 communication rounds in total. We generate both IID and Non-IID data distributions for each dataset. To simulate a realistic Non-IID setup, we utilize a Dirichlet distribution (Minka, 2000) in which the Non-IID degree ($d$) is set to 0.5. It is important to note that smaller values of d indicate more heterogeneous data, and the effect of Non-IID degree is compared for $d = [0.1, 0.3, 0.5, 0.7, 0.9]$.

**Attack Setup.** We perform two non-adaptive and four adaptive poisoning attacks on three standard datasets and evaluate five representative aggregation-based defense methods. The evaluation focuses on three key dimensions: effectiveness, stability, and robustness under adversarial settings. For attack detection, we select representative attacks and assess detection effectiveness, outcomes, and impact on model performance. We report True Positive Rate (TPR) and False Positive Rate (FPR) as primary detection metrics. To assess the impact on model utility, we measure clean accuracy degradation relative to an attack-free baseline.

**Evaluation Metrics.** We evaluated the performance of FLCatcher using the following metrics: True Positive Rate, False Positive Rate, and Accuracy. TPR is the proportion of malicious clients correctly identified as malicious, and FPR is the proportion of benign clients incorrectly classified as malicious. Accuracy (Acc) refers to the global model's classification accuracy on the clean test set, measured as the fraction of correctly predicted test examples. **Compared Methods.** FLCatcher is benchmarked against several representative defenses: *Mkrum*, *Bulyan*, *Trmean*, *Median* and *DnC*. For *Mkrum*, *Bulyan*, *Trmean*, the gradient ratio selected for aggregation is set as $60\%$. For *DnC*, the tolerance rate $\tau$ is tuned to detect the maximum number of malicious clients without degrading model utility.

## 4.2 EXPERIMENTAL RESULTS

We evaluated FLCatcher against multiple defense aggregation schemes under diverse poisoning attacks.

**FLCatcher Outperforms State-of-the-Art Defenses.** *Poisoning attack in FL.* As shown in Figure 2, existing adaptive poisoning attacks severely compromise FL on CIFAR-10, with Min-Max and Min-Sum attacks degrading accuracy by 45.76% and 39.87%, respectively. As shown in Table 2, on simpler datasets such as MNIST and Fashion-MNIST, the impact is less severe due to the relative robustness of the models. Specifically, Min-Max causes accuracy drops of 39.85% and 45.7%, while Min-Sum results in reductions of only 12.34% and 4.01%, respectively.

*Defense Effectiveness of FLCatcher.* Existing defenses are inadequate, causing accuracy drops of up to 47.69% due to their reliance on single-dimensional features, which fail to distinguish between Non-IID variations and malicious perturbations for CIFAR-10. In contrast, FLCatcher incurs only a 2.6% reduction in accuracy by leveraging multi-dimensional gradient analysis. This approach amplifies differences in poisoned gradients across multiple dimensions, enabling earlier and more thorough detection. *Detection Performance of FLCatcher.* As shown in Table 1, our evaluation

| Dataset | Attack Methods | FedAvg | Mkrum | Bulyan | Trmean | Median | Ours |
|---------|----------------|--------|-------|--------|--------|--------|------|
| MNIST | LIE | 96.04 | 93.43 | 81.13 | 96.10 | 91.46 | **96.96** |
| | Fang | 96.25 | 94.52 | 62.64 | 95.88 | 95.33 | **97.04** |
| | Min-Max | 58.28 | 67.51 | 68.18 | 61.67 | 74.21 | **97.14** |
| | Min-Sum | 94.12 | 89.29 | 62.20 | 94.48 | 65.26 | **97.00** |
| Fashion-MNIST | LIE | 85.96 | 82.14 | 68.28 | 86.14 | 76.81 | **88.92** |
| | Fang | 88.49 | 85.63 | 63.60 | 84.94 | 84.46 | **88.96** |
| | Min-Max | 42.69 | 47.16 | 48.76 | 39.29 | 39.73 | **88.51** |
| | Min-Sum | 76.06 | 56.74 | 38.35 | 77.96 | 48.32 | **88.66** |

Table 2: Test accuracy (%) of different aggregation methods under various attacks.

demonstrates that FLCatcher significantly outperforms state-of-the-art defenses in both effectiveness and precision. It achieves an average TPR of 94.47%, with absolute gains of 46.93%, 56.25%, and 55.78% over Mkrum, Bulya, and DnC, respectively. Meanwhile, FLCatchermaintains a low average FPR of 0.72%, markedly outperforming Mkrum, Bulyan, and DnC by margins of 42.4%, 39.72%, and 18.77%. These improvements are attributed to two key components: (i) a Wasserstein distance-based alignment strategy that amplifies subtle malicious gradient patterns, and (ii) an adaptive gap amplification mechanism that distinguishes persistent adversarial behavior from natural Non-IID variations. Furthermore, FLCatcherexhibits strong robustness, maintaining TPR >94.47% consistently across three datasets, two model architectures, and six attack types. In contrast, DnC performs well only on Min-Max attacks (e.g., 99.00% TPR on CIFAR-10) but completely fails on Min-Sum (0% TPR), while Mkrum is only effective on simple attacks or datasets (e.g., 98.11% TPR for Label-Flip on Fashion-MNIST). Notably, FLCatcherachieves near-zero FPR in 10 out of 18 settings, highlighting its precision in minimizing false positives under diverse FL conditions.

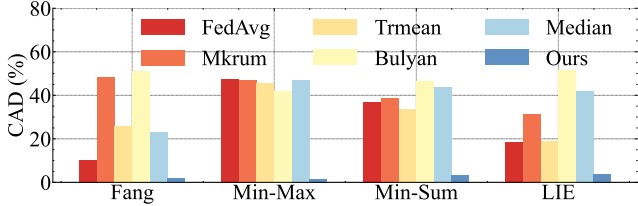

Figure 2: Clean Accuracy Drop (CAD) comparison of different defense methods on CIFAR-10 with AlexNet.

### 4.2.1 ROBUSTNESS EVALUATION OF FLCATCHER.

To further evaluate the performance of FLCatcher, we analyze the impact of Wasserstein distance-based alignment, confirming the robustness of FLCatcher under challenging FL environments.

**Impact of Wasserstein Distance-based Distributional Alignment.** To evaluate how different distance metrics influence FLCatcher, we measured each client's gradient deviation from the me-

dian gradient at every round. Using kernel density estimation, we estimated the probability density distributions of these deviations across various poisoning attacks. Figure 3 illustrates the Min-Sum probability distributions for benign and malicious clients on CIFAR-10 under both IID and Non-IID settings, computed every 30 rounds.

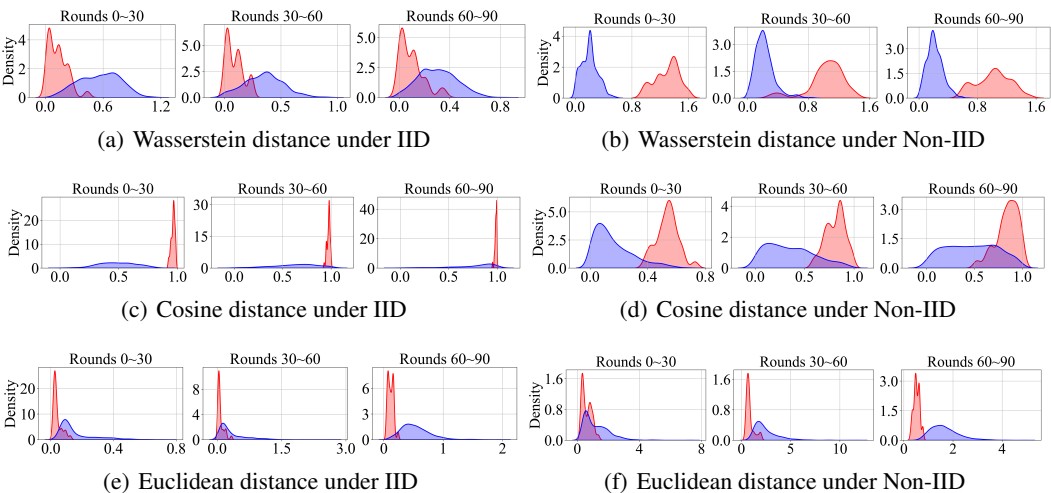

Figure 3: Visualization of gradient distance distributions under different metrics and data settings.

*Under the IID setting*: As shown in Figure 3(c)(e), Euclidean and cosine distances can effectively separate benign and malicious gradients because the client data distributions are similar and homogeneous. This homogeneity leads to consistent gradient directions and magnitudes across benign clients, making deviations caused by malicious updates more pronounced and easier to detect using straightforward geometric measures such as Euclidean distance (which captures magnitude differences) and cosine similarity (which captures directional differences).

*Under the Non-IID setting*: As shown in Figure 3(b), FLCatcher leverages the Wasserstein distance, which offers the clearest separation. This is because Non-IID data increases heterogeneity in the global gradient distribution, obscuring the deviation between benign and malicious updates. Existing adaptive attacks exploit this by crafting poisoned gradients that mimic Non-IID variations in direction (cosine) or magnitude (Euclidean) to evade detection (See Figure 3(d)(f)). In contrast, FLCatcher design *Wasserstein distance-based distributional alignment* to measure differences between entire distributions, capturing richer statistical properties—including higher-order characteristics—that are more difficult for attackers to replicate. Consequently, FLCatcher provides stronger robustness and superior detection capability, even against complex adaptive attack strategies.

**Impact of Varying Non-IID degrees on the defense performance of FLCatcher.** Figure 4 illustrates the TPR and FPR of each method under varying degrees of Non-IID data distribution. As data heterogeneity increases, baseline methods experience significant performance degradation. In contrast, FLCatcher consistently achieves the highest TPR and lowest FPR across all values of $d$, outperforming the other methods. Under the Min-Max attack setting, FLCatcher reaches nearly 100% TPR with 0% FPR. Notably, in extreme Non-IID scenarios, FLCatcher achieves an average TPR improvement of 52.87% on CIFAR-10, demonstrating its superior scalability and robustness across varying levels of data heterogeneity. This superior performance is attributed to FLCatcher's ability to leverage multi-dimensional gradient features and an adaptive compensation mechanism, which together amplify subtle malicious patterns and accurately distinguish them from natural fluctuations caused by heterogeneous data distributions. As a result, FLCatcher maintains effective detection even as Non-IID severity increases, while baselines that rely on simpler or unidimensional detection strategies fail to adapt to the complexities introduced by data heterogeneity.

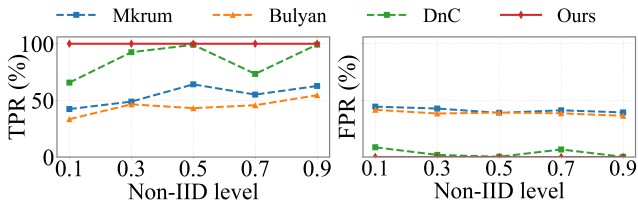

Figure 4: Impact of varying Non-IID degrees on the defense performance of FLCatcher.

**Impact of Percentage of Malicious Client.** We evaluate the robustness of FLCatcher against two representative adaptive attacks—Min-Max and Fang—under varying malicious client ratios (10%-40%). Figure 5 presents the TPR and FPR compared with baseline defenses (Mkrum, Bulyan, DnC).

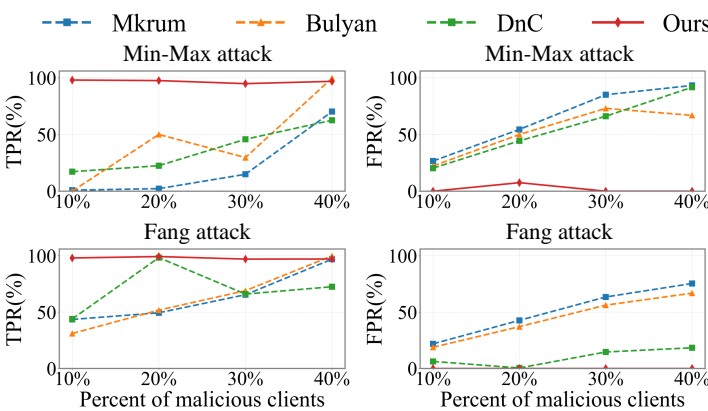

Figure 5: Impact of Malicious Client Ratio on TPR and FPR for CIFAR-10 (AlexNet) under Min-Max and Fang Attacks.

*Under the Min-Max attack*, Mkrum and Bulyan exhibit significantly increasing TPR as the malicious client ratio increases, reaching TPR of up to 96.93% and 99.8% respectively at 40%. However, this comes at the cost of higher FPR, which exceed 66.8% for both methods under the same condition. DnC achieves a relatively high TPR (72.5) at moderate adversary ratios, but its FPR also becomes non-negligible (18.33%). In contrast, FLCatcher consistently achieves near-perfect TPR (96.99%-99.25%) across all malicious client ratios, while maintaining FPR close to zero (<0.25%), demonstrating its robustness against adaptive poisoning attacks.

*Under the Fang attack*, all baseline methods suffer from increased FPR as the attacker ratio grows. For instance, Mkrum and Bulyan reach FPR of 93.22% and 66.86% respectively at 40% adversaries, with only moderate TPR gains. DnC exhibits fluctuating detection accuracy, with TPR and FPR varying unpredictably across ratios. By contrast, FLCatcher not only maintains a high TPR (96.86%) even under 40% malicious clients, but also suppresses the FPR below 7.5%, outperforming all baseline methods in both detection precision and stability. These results confirm that FLCatcher effectively balances high sensitivity with strong specificity across diverse attack strengths, outperforming conventional defenses against adaptive attacks under varying malicious client ratios.

## 5 CONCLUSION

We proposed FLCatcher, a defense for Non-IID FL poisoning attacks that overcomes the limitation of distribution-difference-based defenses by jointly modeling gradient spatial distributions and temporal evolution. Through Wasserstein distance-based alignment and adaptive discriminative gap amplification, FLCatcher robustly distinguishes malicious perturbations from benign heterogeneity. Experiments on standard benchmarks show its average TPR exceeds 94.47% and FPR remains below 0.72%, outperforming state-of-the-art defenses under diverse adaptive attacks.

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
