# OpenReview forum: "FLCatcher: Fingerprinting Poisoning Attack in Non-IID Federated Learning"
_ICLR.cc/2026/Conference — ICLR 2026 Conference Desk Rejected Submission_

### Official Review · Reviewer_LjYQ · 2025-10-16

**Soundness:** 1
**Presentation:** 2
**Contribution:** 1
**Rating:** 2
**Confidence:** 4

**Summary:**

This paper introduces FLCatcher, a defense framework designed to mitigate poisoning attacks in Non-IID federated learning (FL). Unlike existing defenses that rely primarily on inter-client distribution differences, FLCatcher jointly models the spatial distribution and temporal evolution of gradients to more effectively capture malicious updates. By employing Wasserstein distance-based alignment and adaptive discriminative gap amplification, the method enhances the separation between benign and malicious clients even under heterogeneous conditions.

**Strengths:**

1. The proposed method is easy to understand.
2. The writing is good.

**Weaknesses:**

1. For a paper targeting ICLR 2026, all compared defense methods are from 2021 or earlier. This suggests that the authors may not be fully aware of more recent advances in federated backdoor defense. Incorporating and discussing state-of-the-art methods would provide a fairer and more convincing comparison.

2. The experimental evaluation is confined to small-scale datasets and lightweight vision models. There is no assessment on larger benchmarks such as ImageNet or with more recent architectures like Vision Transformers (ViT). Moreover, the study does not extend beyond computer vision tasks—for example, no experiments are conducted on NLP datasets. As a result, the generality and scalability of the proposed method remain unclear.

3. Prior work (e.g., Back to the Drawing Board: A Critical Evaluation of Poisoning Attacks on Production Federated Learning) has shown that in realistic industrial settings, the proportion of malicious clients is often below 1%, and in some cases as low as 0.01%. Under such conditions, even simple defenses such as norm clipping are often sufficient to protect the system. Therefore, the practical significance and industrial relevance of this work are somewhat limited.

4. The paper does not provide any theoretical analysis or formal guarantees regarding the effectiveness or convergence of the proposed defense. Without such analysis, it is difficult to assess whether the observed empirical robustness holds under broader or more general conditions.

5. The paper does not evaluate  the proposed method under AGR-tailor attacks. Although the authors mention “adaptive poisoning attacks”, those are designed for other defenses rather than being specifically tailored to the proposed approach. A proper adaptive attack evaluation is essential to validate the robustness claims.

6. Several important experimental components are missing, including ablation studies and hyperparameter sensitivity analyses. These evaluations are necessary to understand the contribution of individual components and the stability of the method under different settings.

**Questions:**

Please refer the the Weaknesses.

---

### Official Review · Reviewer_g8He · 2025-10-27

**Soundness:** 3
**Presentation:** 2
**Contribution:** 2
**Rating:** 4
**Confidence:** 4

**Summary:**

The paper addresses the challenge of detecting model poisoning attacks in federated learning (FL) under non-IID client data distributions, where benign client gradients naturally vary. This intrinsic heterogeneity renders existing gradient-distribution-based defenses ineffective. To overcome this, the authors propose FLCatcher, a defense mechanism combining: (1) Wasserstein distance-based gradient distribution alignment to capture subtle inter-layer deviations. (2) Adaptive discriminative gap amplification through temporal trajectory analysis, leveraging gradient evolution to distinguish benign vs. malicious behavior. (3) A federated behavior embedding strategy that integrates distributional (spatial) and temporal features. Experimental results across MNIST, CIFAR-10, and Fashion-MNIST demonstrate superior detection performance, achieving average TPR > 94.47% and FPR < 0.72%, outperforming state-of-the-art defenses such as Krum, Bulyan, Median, and DnC.

**Strengths:**

1. Clear motivation. The paper convincingly identifies a critical gap: existing defenses rely on static gradient similarity detection, which is unreliable under Non-IID settings. It emphasizes realistic attacker strategies (adaptive poisoning, slow drift, camouflage using statistical heterogeneity).
2. Strong empirical results. Comprehensive evaluation with various datasets, architectures, and attacks. Demonstrates robustness under increased Non-IID severity and higher proportions of malicious clients.
3. Good practical relevance. Addresses real-world FL constraints: non-IID distributions, adaptive attacks, temporal evolution of behavior.

**Weaknesses:**

1. Missing defense baselines。 While backdoor attacks targeting Non-IID FL systems are considered a persistent and significant threat, several existing works have already proposed effective defense mechanisms. The authors should include these state-of-the-art defense baselines, such as [1], [2], and [3], to strengthen the claim that FLCatcher outperforms current defenses.

2. Missing attack baselines。 The authors should also evaluate FLCatcher against more advanced backdoor attack algorithms that have been empirically shown to bypass statistical defense strategies, including [4]. This is necessary to demonstrate the robustness of the proposed method under realistic and adaptive threat models.

3. Potential computational overhead。 The proposed use of Wasserstein distance requires computing distributional shifts across client gradients. This may be acceptable for small models such as AlexNet; however, for larger architectures like ResNet-18 or ResNet-34, the computational cost could increase significantly. This raises concerns regarding the scalability and practical deployment of the method in real-world federated learning systems.

[1] Nguyen, Thien Duc, et al. "{FLAME}: Taming backdoors in federated learning." 31st USENIX Security Symposium (USENIX Security 22). 2022.
[2] Li, Songze, and Yanbo Dai. "{BackdoorIndicator}: Leveraging {OOD} Data for Proactive Backdoor Detection in Federated Learning." 33rd USENIX Security Symposium (USENIX Security 24). 2024.
[3] Fereidooni, Hossein, et al. "Freqfed: A frequency analysis-based approach for mitigating poisoning attacks in federated learning." arXiv preprint arXiv:2312.04432 (2023).
[4] Zhang, Zhengming, et al. "Neurotoxin: Durable backdoors in federated learning." International conference on machine learning. PMLR, 2022.

**Questions:**

1. Could authors provide extra empirical results when evaluating FLCatcher on more advanced backdoor attack algorithms, and in comparison with defense mechanisms designed speicificly for Non-IID scenarios?
2. Could authors justify the scalability of FLCatcher?

---

### Official Review · Reviewer_i3F2 · 2025-10-31

**Soundness:** 2
**Presentation:** 3
**Contribution:** 2
**Rating:** 4
**Confidence:** 4

**Summary:**

The paper tackles poisoning detection in federated learning (FL) when client data are non‑IID, arguing that benign updates naturally look diverse and can camouflage attackers. The authors proposed FLCatcher which combines two ideas: it measures layer‑wise distributional differences between a client’s gradient and a reference using Wasserstein distance, and it models how each client’s updates evolve over time with an exponential moving average, then clusters clients and applies a dynamic threshold which only flags abnormal clients. Experiments report strong results against both non‑adaptive and adaptive attacks.

**Strengths:**

- The work is well‑motivated by the core failure mode of many defenses under heterogeneity and squarely focuses on that setting.
- Experiments conducted on a broad range of attack groups across three datasets.
- The authors present an intuitive separation story for the metric choice: Wasserstein continues to separate benign from malicious under non‑IID, where cosine and Euclidean blur, supported by distribution plots.

**Weaknesses:**

- How gradients become distributions is unspecified (support construction, weighting scheme, ground distance), but these choices strongly affect both accuracy and runtime. Concrete definitions and ablations are needed to justify this component
- The method involves computing per-layer optimal-transport distances for each client in every training round, this step could be computationally intensive in large-scale federated systems, but the paper does not include any analysis of computational or communication overhead.
- Strong, relevant baselines discussed in the introduction (RFA [1], FoolsGold [2]) do not appear in the detection table and should be added.
- Training details are incomplete (for example, the learning‑rate decay “Line 298 - at a rate of c” leaves c undefined, local epochs per round are not stated).

[1] Pillutla, Krishna, Sham M. Kakade, and Zaid Harchaoui. "Robust aggregation for federated learning." IEEE Transactions on Signal Processing 70 (2022): 1142-1154.

[2] Fung, Clement, Chris JM Yoon, and Ivan Beschastnikh. "The limitations of federated learning in sybil settings." 23rd International symposium on research in attacks, intrusions and defenses (RAID 2020). 2020.

**Questions:**

- What is the justification behind using a non-standard “60% gradient ratio” configuration for baselines such as Mkrum, Bulyan, and Trmean, and how might this choice affect their reported false-positive rates?
- Several important quantities, such as the learning-rate decay constant and local epochs, are undefined; could the authors clarify these details to ensure full reproducibility?
- Have the authors evaluated how the method behaves under partial client participation or on larger and non-vision datasets, and if not, could this omission limit the generality of the findings?

---

### Official Review · Reviewer_8ci4 · 2025-10-31

**Soundness:** 2
**Presentation:** 1
**Contribution:** 2
**Rating:** 4
**Confidence:** 4

**Summary:**

This paper proposes FLCatcher, a defense framework for poisoning attacks in Non-IID Federated Learning that jointly models the spatial distribution and temporal consistency of client gradients. It combines a Wasserstein distance–based alignment strategy with an adaptive discriminative gap amplification mechanism to distinguish malicious from benign updates. Extensive experiments on multiple datasets and attacks demonstrate that FLCatcher achieves high detection accuracy and robustness, significantly outperforming existing defenses under challenging Non-IID conditions.

**Strengths:**

1. The paper introduces a new perspective by modeling client behaviors jointly through spatial distribution alignment (via Wasserstein distance) and consider temperal relationship between each client itself. This design is conceptually novel compared to prior single-round or purely distance-based defenses.

2. The experiments are extensive and systematically cover both IID and Non-IID settings, multiple datasets (CIFAR-10, MNIST, Fashion-MNIST), and diverse poisoning attacks (LIE, Fang, Min-Max, Min-Sum, FedGhost). The consistent improvements in TPR and low FPR across all settings provide convincing empirical support for the method’s robustness.

3. The visualizations are clear and effective. In particular, Figure 3 comparing Wasserstein, cosine, and Euclidean distances under IID and Non-IID scenarios helps illustrate *why* Wasserstein-based alignment yields better separation between benign and malicious gradients, directly supporting the paper’s core motivation.

**Weaknesses:**

1. Important: The methodology of this paper needs a heavy revision. Many symbols are introduced without a clear order of definition/explanation such as $C_b, C_s, μ_i^{(l)}, D_S, α_l, etc.$. It’s unclear how $\tau$ is calculated or updated, and reusing $\tau$ (also the tolerance rate in DnC) makes the notation overloaded and confusing. In §3.1.1, the sentence ``then cluster benign clients and suspicious clients using DBSCAN to capture semantic consistency and differentiate truly malicious updates from benign extremes'' is hard to understand: what features are clustered, how are $C_s$ (malicious) clusters determined, how is non-overlap enforced so a cluster contains only benign or only malicious clients, and how are clusters labeled in an unsupervised step?

The intuition and explanation for Algorithm 1 is missing. Why set $\Delta_1$ to the gap between "top suspicious" and "benign max" before we can reliably identify benign groups (risk of circular logic/misclassification)? What is $R$ (meaning, initialization, update rationale)? Line 5 is confusing: it seems only the single $C_i$ with the highest $L_i$ is added to the malicious set; is that really what the authors mean? Also, how is the $\tau$ from (iii) *Adaptive Threshold* actually used here (it doesn’t appear in the pseudo-code)? Overall, the core detection logic is under-specified and not reproducible. What is $L_k$ in Algortihm 1?

2. Does the claim that "malicious clients … produce higher deviations from the EMA-predicted gradients" hold in most cases? For example, what happens if an adaptive adversary could submit low-variance, thereby collapsing the temporal score and evading detection. Please provide empirical results or discuss more about this adaptive attack.

3. Batchsize = 125 seems a little weird for the experimental setting.

4. The experiment section misses some important ablation study. (i) The clean accuracy is not reported. (ii) OT/Wasserstein distances are expensive at scale, but it's still unclear why the WD is necessary but not other distances (quantitatively). (iii) There is no controlled ablation that isolates the contributions of each component in the method. Show TPR/FPR with each component removed and combinations thereof. (iv) Other clustering methods are not studied.

5. Latest defenses against poisoning attacks in FL have not been compared.

**Questions:**

Overall, the methodology seems really confusing, which raises concern about the soundness of this work. Given the current version, it's hard to say which part contributes the most to the novelty of this work.

---

### Note · Program_Chairs · 2026-01-17
**Submission Desk Rejected by Program Chairs**

The following references in this submission do not refer to real documents and/or have major errors in bibliographic information:

 Diego Martínez Beltrán, J. González, P. Dutta, and S. Chatterjee. Communication security challenges and solutions in decentralized federated learning. arXiv preprint arXiv:2307.11730, 2023